# Optimal LED Wavelength Composition for the Production of High-Quality Watermelon and Interspecific Squash Seedlings Used for Grafting

**Filippos Bantis [1,\*], Athanasios Koukounaras [1] , Anastasios S. Siomos [1], Kalliopi Radoglou [2] and Christodoulos Dangitsis [3]**

1   Department of Horticulture, Aristotle University, 54124 Thessaloniki, Greece; thankou@agro.auth.gr (A.K.); siomos@agro.auth.gr (A.S.S.)
2   Department of Forestry and Management of the Environment and Natural Resources Democritus University of Thrace, 68200 Nea Orestiada, Greece; kradoglou@fmenr.duth.gr
3   Agris S.A., Kleidi, 59300 Imathia, Greece; cdaggitsis@agris.gr
*   Correspondence: fbantis@agro.auth.gr; Tel.: +30-2310-994123

**Abstract:** Watermelon is cultivated worldwide and is mainly grafted onto interspecific squash rootstocks. Light-emitting diodes (LEDs) can be implemented as light sources during indoor production of both species and their spectral quality is of great importance. The objective of the present study was to determine the optimal emission of LEDs with wide wavelength for the production of watermelon and interspecific squash seedlings in a growth chamber. Conditions were set at 22/20 °C temperature (day/night), 16 h photoperiod, and $85 \pm 5\ \mu mol\ m^{-2}\ s^{-1}$ photosynthetic photon flux density. Illumination was provided by fluorescent (FL, T0) lamps or four LEDs (T1, T2, T3, and T4) emitting varying wide spectra. Watermelon seedlings had greater shoot length, stem diameter, cotyledon area, shoot dry weight-to-length (DW/L) ratio, and Dickson's quality index (DQI) under T1 and T3, while leaf area and shoot dry weight (DW) had higher values under T1. Interspecific squash seedlings had greater stem diameter, and shoot and root DW under T1 and T3, while leaf and cotyledon areas were favored under T1. In both species, T0 showed inferior development. It could be concluded that a light source with high red emission, relatively low blue emission, and a red:far-red ratio of about 3 units seems ideal for the production of high-quality watermelon (scion) and interspecific squash (rootstock) seedlings.

**Keywords:** *Citrullus lanatus*; *Cucurbita maxima* × *C. moschata*; TZ-148; scion; rootstock; light-emitting diodes; photomorphogenesis; controlled environment agriculture; growth chamber

## 1. Introduction

Watermelon (*Citrullus lanatus*) is an economically important crop worldwide producing over 100 million tons annually (2010–2017, FAOSTAT Database). Watermelon crop is mainly established using grafted seedlings consisting of two plant segments: the scion and rootstock. The scion segment is a watermelon hybrid offering a number of desired traits including relatively high yield as well as qualitative characteristics such as pleasant fruit flavor and aroma, and higher antioxidant compound content. The rootstock segment provides the grafted seedling with greater protection against environmental stress factors (heavy metals, salinity, low temperatures, etc.) [1,2] and soil-borne pathogens [3,4]. TZ-148 (interspecific squash, *Cucurbita maxima* × *C. moschata*) is the most commonly used rootstock for watermelon grafting throughout the world. In Korea and Japan, more than 90% of watermelon seedlings were grafted in 2005, exceeding the production of 300 million grafted seedlings, while in China the percentage of grafted watermelon seedlings was 20%. In Europe, Spain and Italy

produced about 60 million grafted watermelon seedlings in 2009 [1]. In Greece about 30 million grafted vegetable seedlings are produced annually, 19 million (about 63%) of which are watermelon seedlings (Th. Koufakis, Agris S.A., Kleidi, Imathia, Greece, personal communication). Moreover, watermelon shows a high yield comparable to tomatoes, while the production in Greece constitutes over 20% of the annual production in the European Union (2010–2017, FAOSTAT Database). In the northern hemisphere grafted watermelon seedlings are transplanted from early February to April, and therefore watermelon grafting is performed between December and April. At that time, in countries with a latitude of 30° to 45° such as Japan, Korea, Italy, Spain, and Greece, the natural daylength is between 9 (December) and 13 (April) hours which is not sufficient for the production of Cucurbit seedlings. Moreover, during that period the weather is typically colder and cloudier, thus, light intensity is generally insufficient for optimal plant development. Seedling nurseries annually face the above environmental constraints which lead to seedlings with inferior growth, as well as a lack of uniformity. Therefore, there is a growing need for increased light intensity in order for nurseries to provide the growers with high-quality seedlings even during winter months with the purpose of achieving production earliness. In the past, watermelon seedling quality had a vague meaning that included terms such as plant uniformity, proper size, thick stem, and a well-developed root system, among others. Two recently published articles [5,6] offer a proper description of the quality of to-be-grafted watermelon seedlings as well as grafted seedlings using a number of objective measurements.

For several decades supplementary light sources have been used for the purpose of increasing the photoperiod and/or the photosynthetic photon flux density (PPFD). Light sources such as incandescent, fluorescent (FL), metal halide, and high pressure sodium (HPS) are implemented in controlled environment agriculture (CEA) such as greenhouses and growth chambers [7]. CEA is an excellent alternative for nursery production since it enables the planning of production ahead of time, limits the impact of external constraints, and generally reduces uncertain factors. The above-mentioned light sources have the ability to enhance the quantitative and qualitative yield of the seedlings but they exhibit certain limitations [8]. Light-emitting diodes (LEDs) are solid-state light sources offering a number of economical and cultivation advantages compared to conventional light sources. The greatest benefits of LEDs are precise spectral and intensity control, high luminous efficacy, low energy requirements, high lifespan, and low heat emission [9,10]. Nowadays, LEDs are widely employed for CEA since their potential to enhance plant growth and development has been tested extensively [11]. Plants utilize light not only as energy that drives photosynthesis, but also as information that directs their growth and development [8]. Quality, intensity, duration, and direction are the most essential light features that control several developmental parameters of plants [12,13]. Incident light is mainly harvested by chlorophylls a and b in the red (R; 665 and 642 nm, respectively) and blue (B; 430 and 453 nm, respectively) wavelengths, while other pigments such as anthocyanins and carotenoids contribute to light-absorbance in a wide spectral range [14]. Moreover, plants are equipped with protein photoreceptors (i.e., phytochromes, cryptochromes, phototropins, zeitlupe family, UVR8) that perceive, interpret, and transduce incident light signals and acting through gene expressions they regulate changes in the whole plant level [15]. These photoreceptors are important for several plant responses such as germination, stem elongation, flowering, shade-avoidance response, phototropism, stomatal opening, and photomorphogenesis among others [12,15–19]. The responses above are mainly associated to far-red (FR), R, and B wavelengths, as well as their respective ratios (i.e., R/FR and R/B). However, distinct responses have also been attributed to green (G) wavelength and ratios such as B/G and R/G [20] since G light reaches deeper in the canopy and is utilized for photosynthesis and photomorphogenesis [21]. Moreover, UV light has been known to promote plant compactness but also to decrease photosynthetic activity [22].

To our knowledge, no research results have been published about watermelon and squash seedlings grown under the influence of varying radiation spectra. On the contrary, another economically important cucurbit, cucumber, showed accelerated growth under supplemental green (G) or orange lights [23]. Also working with cucumber seedlings, Hernandez and Kubota [24] found decreased

growth rate (i.e., dry mass) and morphological (i.e., stem extension and leaf area) parameters under increasing B light when both B and R were present, while Snowden et al. [24] reported dry mass reduction when B light increased from 11% to 28% under 500 µmol m$^{-2}$ s$^{-1}$. However, there is a strong worldwide tendency from producers to use LEDs with a wide wavelength in order to improve the working environment for employees.

The quality and cost of cucurbit seedlings' production has high potential to benefit from indoor cultivation under artificial lighting. The objective of our research was to identify the optimal spectral emission of LEDs with wide wavelength for the production of high-quality watermelon scions and interspecific squash rootstocks in a growth chamber. In parallel, the two economically valuable cucurbit species were studied with respect to their response under varying irradiation wavelengths.

## 2. Materials and Methods

### 2.1. Plant Material and Germination

The experiments were conducted at the facilities of the Forest Research Institute, Vasilika, Thessaloniki, Greece, and all measurements were performed at the Lab of Vegetable Crops of the Aristotle University of Thessaloniki.

Watermelon (*Citrullus lanatus*) "Celine F1" (HM.Clause SA, Portes-Les-Valence, France) or interspecific squash rootstock (*Cucurbita maxima × C. moschata*) "TZ-148" (HM.Clause SA, Portes-Les-Valence, France) seeds were sown in plastic 171 and 128-cell plug trays (both: 67 × 33 cm, G.K. Rizakos S.A., Lamia, Greece), respectively. The plastic plug trays were filled with a 5:1:2 mixture of peat, perlite, and vermiculite, and after sowing they were moved into a custom-built growth chamber (25 °C temperature, 95%–98% relative humidity, and darkness) for 72 (watermelon) or 48 (interspecific squash) h until germination.

### 2.2. Growth Chamber and Light Conditions

Upon germination, three plug trays per light treatment were moved in another custom-built growth chamber (0.6 × 1.2 × 0.55 m per shelf, 0.5 m lamp–plant distance) with controlled conditions set at 22 °C/20 °C day/night temperature and 80 ± 10% relative humidity. Irrigations were applied with water sprinklers.

Illumination was provided by four LEDs (120 cm, 132 W) indicated as T1, T2, T3, and T4, emitting wide spectra with varying far-red (FR), R, G, B, and ultraviolet (UV) radiation percentages. Fluorescent lamps (T0; Osram, Fluora, Munich, Germany) were used as the control treatment. Spectral parameters such as light distribution, blue and red peak wavelengths, red:blue (R:B) ratio, red:far-red (R:FR) ratio, phytochrome photostationary state (PSS) (calculated following Sager et al. [25]), correlated color temperature (CCT), and color rendering index (CRI) were obtained using a spectroradiometer (HD 30.1 spectroradiometer, DeltaOhm Srl, Padova, Italy) and are presented in Table 1. The photoperiod was set at 16 h and photosynthetic photon flux density (PPFD) was maintained at 85 ± 5 µmol m$^{-2}$ s$^{-1}$ emitted by two lamps per light treatment. In a recent study, the above-mentioned PPFD proved beneficial for the growth of watermelon and interspecific squash seedlings before grafting [26]. Each species was tested independently in two separate experiments which were performed twice. Similar conclusions were reached from each experiment and therefore only results from the first experiment are presented.

### 2.3. Sampling and Measurements

Sampling and measurements were performed after 13 and 12 days of illumination for watermelon and interspecific squash, respectively. Watermelon seedlings were cultivated for one more day compared to interspecific squash seedlings in order to achieve similar stem diameter for efficient grafting. For each species, 12 evenly distributed seedlings were sampled per tray reaching a total of 36 seedlings per species and light treatment.

Depending on their appearance at the end of the experimental period, seedlings were grouped by experienced personnel to quality categories namely "lowest not acceptable", "lowest acceptable",

"optimum", and "highest acceptable" established by Bantis et al. [5]. Regarding morphological and physiological parameters, valuable seedling quality indicators are the root system biomass, the area of true leaves and cotyledons, and the cotyledon color [5]. Shoot length, stem diameter, and cotyledon thickness were measured with a digital caliper (Powerfix, Milomex, Pulloxhill, UK), while the area of true leaves or cotyledons were determined with a leaf area meter (LI-3000C, LI-COR biosciences, Lincoln, NE, USA). Leaf chlorophyll content was determined using a chlorophyll meter (CCM-200 plus, Opti-Sciences, Hudson, NH, USA). In addition, both watermelon and interspecific squash cotyledon color were determined from colorimetric coordinates obtained using a digital colorimeter (CR-400 Chroma Meter, Konica Minolta Inc., Tokyo, Japan). Colorimetric parameters obtained were lightness (L*), chroma (C*), hue (h○), and a*/b* ratio (a*: red/green coordinate; b*: yellow/blue coordinate) [27]. Dry weight (DW) of shoots and roots was recorded, while their dry root-to-shoot ratio (R/S) was also calculated. Dry weight was obtained after three days of drying in an oven at 72 °C. Moreover, shoot dry weight-to-length (DW/L) ratio and Dickson's quality index (DQI) were also calculated from the above-mentioned values. DQI was calculated as follows [28]:

$$\text{Quality index} = \frac{\text{Seedling total dry weight (g)}}{\frac{\text{Height (mm)}}{\text{Stem diameter (mm)}} + \frac{\text{Shoot dry weight(g)}}{\text{Root dry weight (g)}}} \tag{1}$$

### 2.4. Statistical Analysis

IBM SPSS software (SPSS 23.0, IBM Corp., Armonk, NY, USA) was employed for statistical analysis. Specifically, analysis of variance (ANOVA) was used for data analysis since we examined five different light treatments. Each mean value was computed from $n = 36$ seedlings except for root dry weight, R/S ratio, and DQI which were computed from $n = 18$ seedlings. Mean comparisons were performed by Tukey's test at a significance level of $p = 0.05$.

**Table 1.** Spectral distribution, blue and red peak wavelengths, red:blue (R:B) ratio, red:far-red (R:FR) ratio, phytochrome photostationary state (PSS), correlated color temperature (CCT), and color rendering index (CRI) for the light treatments tested.

| Parameters | Light Treatment | | | | |
|---|---|---|---|---|---|
| | T0 | T1 | T2 | T3 | T4 |
| UV %; 380–399 nm | 0.11 | 0.02 | 0.02 | 0.02 | 0.36 |
| Blue %; 400–499 nm | 20.04 | 7.62 | 10.90 | 11.38 | 20.59 |
| Green %; 500–599 nm | 40.97 | 2.34 | 18.54 | 13.85 | 36.46 |
| Red %; 600–699 nm | 34.75 | 67.25 | 62.20 | 56.48 | 36.92 |
| Far-red %; 700–780 nm | 4.12 | 22.77 | 8.34 | 18.28 | 5.68 |
| Blue peak wavelength (nm) | 436 | 448 | 448 | 448 | 461 |
| Red peak wavelength (nm) | 612 | 660 | 631 | 660 | 660 |
| R:B | 0.63 | 10.12 | 5.58 | 6.39 | 1.93 |
| R:FR | 1.98 | 2.69 | 5.30 | 2.68 | 5.80 |
| PSS | 0.89 | 0.75 | 0.88 | 0.73 | 0.87 |
| CCT (K) | 3830 | 0 | 1624 | 2143 | 5034 |
| CRI | 83.5 | 0 | 66.1 | 71.0 | 87.7 |

## 3. Results

The normal seedling development was verified by the absence of visual irregularities during seedling growth (Figure 1). Seedlings were uniformly developed leading to low standard error values in all parameters and treatments. In both species, T0, T2, and T4 did not induce the production of any "optimum" seedlings up to the day of sampling, while T1 promoted the development of the most "optimum" characterized seedlings. Specifically, watermelon seedlings grown under T1 were characterized as 28% "optimum", while T3 only developed 52% "lowest acceptable" seedlings. Seedlings of the remaining treatments were characterized as 100% "lowest not acceptable". Quite

similarly in interspecific squash, seedlings of T1 were 39% "optimum", while seedlings of T3 were 11% "optimum". T2, T4, and T0 only developed 37%, 24%, and 12% "lowest acceptable" seedlings, respectively (Figure 2).

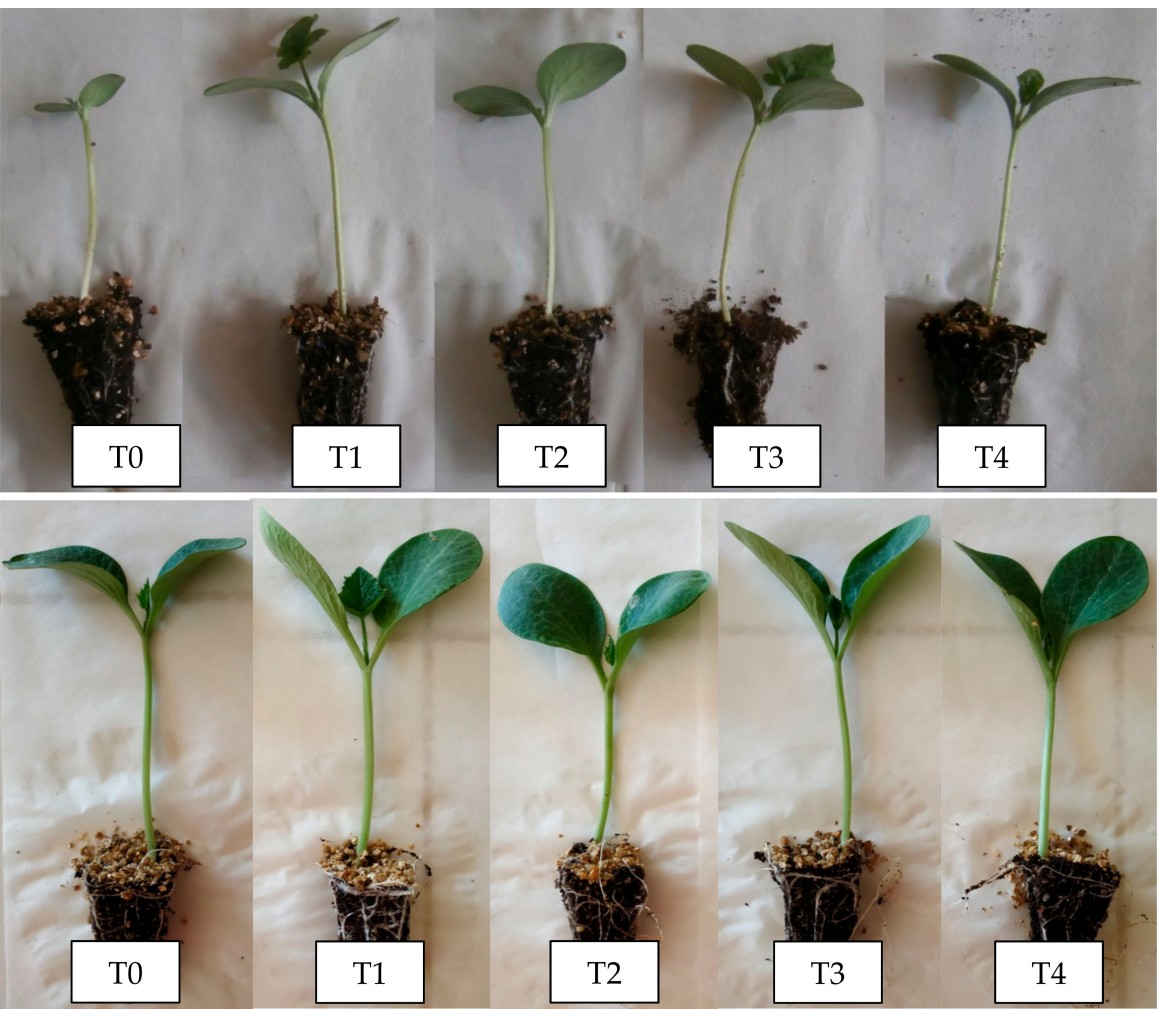

**Figure 1.** Watermelon (top) and interspecific squash (bottom) seedlings after 13 or 12 days, respectively, in a growth chamber under five different light treatments described in Table 1.

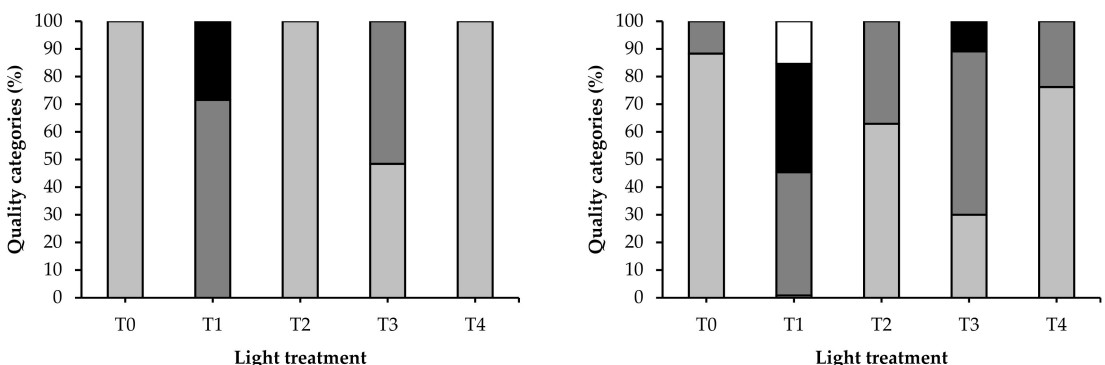

**Figure 2.** Quality categories (%) of watermelon (left) and interspecific squash (right) seedlings after 13 or 12 days, respectively, in a growth chamber under five different light treatments described in Table 1. ▫: Lowest not acceptable; ▪: lowest acceptable; ■: optimum; □: highest acceptable. Each percentage was computed from three trays each having $n = 108$ or 80 seedlings for watermelon and interspecific squash, respectively.

Regarding watermelon color determination, differences between the light treatments could be distinguished by eye. Seedlings grown under T0 developed darker (higher L*) leaves compared to T2, T3, and T4. In addition, T1 seedlings had more intense color (greater C*), while h° and a*/b* ratio were also significantly affected by the different light treatments (Table 2). Watermelon seedlings developed significantly higher and thicker stems under T1 (+19% and +35% respectively, compared to T0) and T3 (+12% and +36% respectively, compared to T0) compared to the rest of the treatments, but cotyledon thickness was not significantly affected by the different irradiation spectra (Figure 3). Leaves and cotyledons were significantly more expanded under T1 (+790% and +82%, respectively) and T3 (+608% and +85%, respectively) compared to T0 and the other LEDs, while T0 imposed the least effect on leaf and cotyledon area development compared to the rest of the treatments (Figure 3). In addition, T1 again formed significantly greater DW (+339% compared to T0). Root dry weight was by far the greatest under all LEDs (between +92% and +139%) compared to T0 (Figure 3). However, R/S ratio was significantly greater under T0 compared to all LEDs (Figure 3). Moreover, leaf chlorophyll content was significantly greater under T0, T2, and T4, compared to T1 and T3. Finally, two quality parameters, DW/L and DQI, revealed significant differences between T1 and T3 (highest values), and T0, T2, and T4 (lowest values) (Figure 3).

Regarding the color evaluation of interspecific squash, T3 and T1 demonstrated darker (higher L*) and more intense (higher C*) color, while h° and a*/b* also exhibited significant differences among the light treatments (Table 2). Interspecific squash seedlings grown under T2 and T4 developed stems with the least length and diameter compared to all the other treatments. Moreover, T0 and T2 led to seedlings with the least cotyledon thickness (Figure 3). T1 led to significantly greater area of true leaves (+139% compared to T0), while T1 and T3 exhibited greater cotyledon area (+39% and +30% respectively, compared to T0) (Figure 3). Dry weight of interspecific squash shoots was significantly greater under T1 (+28% compared to T0) and T3 (+35% compared to T0). Moreover, root DW was positively affected by T3 and T1 (+64% and +60% respectively, compared to T0), while R/S ratio was not affected by the different light treatments (Figure 3). Leaf chlorophyll content was greater under T0 compared to all LEDs (Figure 3). In addition, DW/L and DQI parameters were found lower under T0 compared to all LED treatments (Figure 3).

**Table 2.** Colorimetric parameters of watermelon and interspecific squash seedlings after 13 or 12 days, respectively, in a growth chamber under five different light treatments described in Table 1. L*: lighting; C*: chroma; h°: hue angle; a*: red/green coordinate; b*: yellow/blue coordinate. Mean values (± SE), within a row, followed by different letters are significantly different (*p* ≤ 0.05). Each mean value was computed from *n* = 36 seedlings.

| Species | Parameters | Light Treatment | | | | |
|---|---|---|---|---|---|---|
| | | T0 | T1 | T2 | T3 | T4 |
| **Watermelon** | **L*** | 43.33 ± 0.41 a | 42.60 ± 0.26 ab | 40.94 ± 0.33 c | 42.09 ± 0.57 bc | 42.69 ± 0.98 bc |
| | **C*** | 21.15 ± 0.33 c | 25.31 ± 0.51 a | 20.45 ± 0.44 c | 22.69 ± 0.54 b | 19.82 ± 0.75 c |
| | **h°** | 127.0 ± 0.27 b | 126.6 ± 0.21 b | 128.6 ± 0.22 a | 127.7 ± 0.16 b | 128.8 ± 0.28 a |
| | **a*/b*** | −0.76 ± 0.01 a | −0.74 ± 0.01 a | −0.80 ± 0.01 b | −0.77 ± 0.00 a | −0.81 ± 0.01 b |
| **Interspecific Squash** | **L*** | 38.27 ± 0.45 d | 42.05 ± 0.32 ab | 40.15 ± 0.28 c | 42.48 ± 0.40 a | 40.96 ± 0.30 bc |
| | **C*** | 20.81 ± 0.41 c | 26.86 ± 0.35 a | 24.34 ± 0.40 b | 27.61 ± 0.41 a | 24.68 ± 0.32 b |
| | **h°** | 129.5 ± 0.22 a | 127.0 ± 0.15 cd | 128.0 ± 0.17 b | 126.7 ± 0.19 d | 127.6 ± 0.12 bc |
| | **a*/b*** | −0.83 ± 0.01 d | −0.75 ± 0.00 ab | −0.78 ± 0.00 c | −0.75 ± 0.01 a | −0.77 ± 0.00 bc |

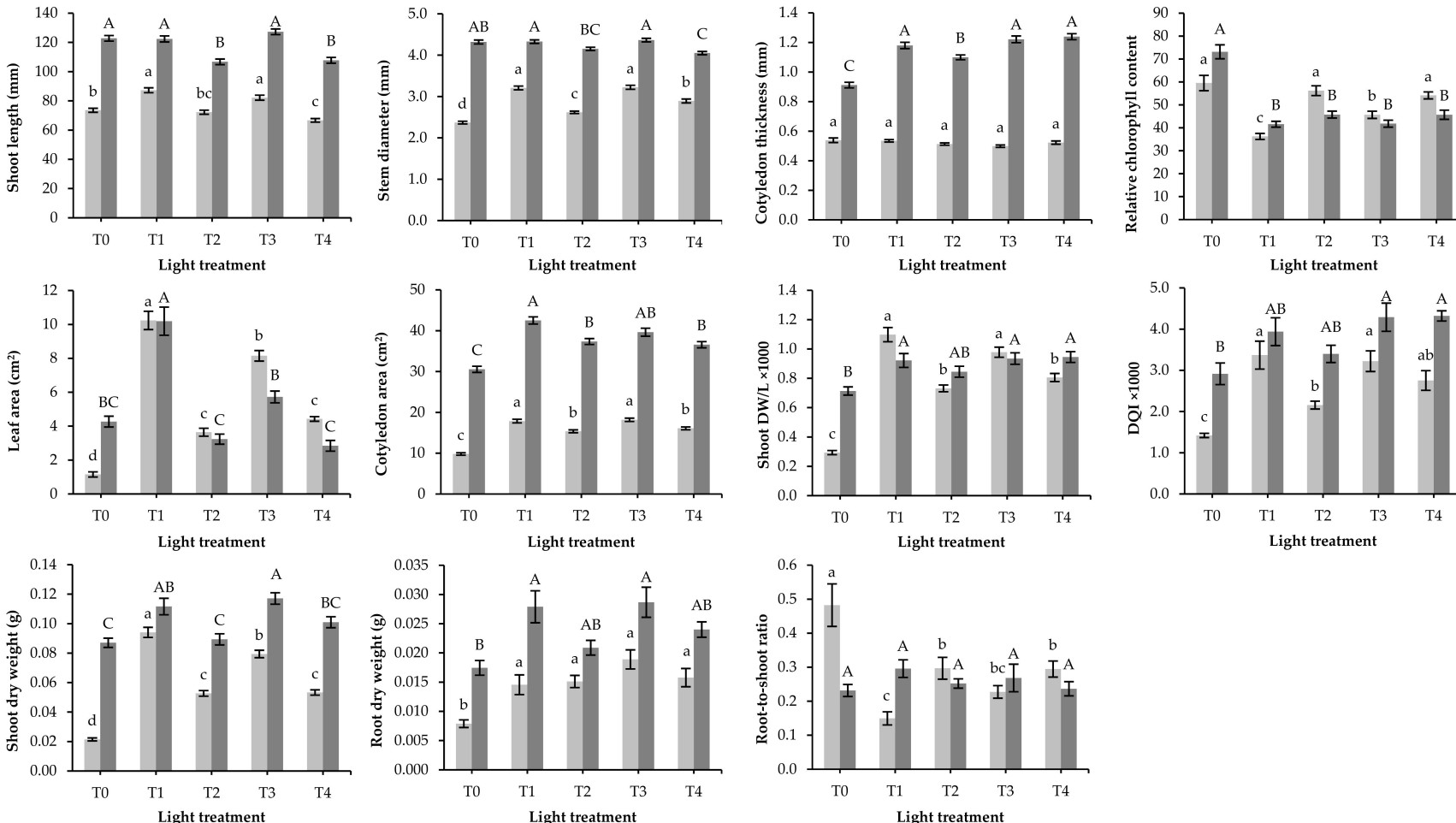

**Figure 3.** Morphological, growth, and developmental parameters of watermelon (□) and interspecific squash (■) seedlings after 13 or 12 days, respectively, in a growth chamber under five different light treatments described in Table 1. DW/L: shoot dry weight-to-length ratio; DQI: Dickson's quality index. Bars followed by different letters are significantly different ($p \leq 0.05$). Each mean value was computed from $n = 36$ seedlings except for root dry weight, root-to-shoot ratio, and DQI which were computed from $n = 18$ seedlings.

## 4. Discussion

The nursery industry faces the challenge of the production of high-quality products (seedlings) under non-optimal environmental conditions. Therefore, there is a strong tendency to direct seedling production towards entirely protected environments (i.e., indoors production where artificial lighting is necessary), especially for high economic value products such as grafted seedlings. Light acting through its wavelength imposes significant responses on plants, which are often highly species-dependent. By controlling light wavelength, we have the potential to manipulate seedling production according to the needs of the nursery industry (e.g., shorter growth cycle) and the market (e.g., production of compact seedlings). Our study aimed to provide additional input on the understanding of the effects of light quality on two important cucurbits used for grafting, watermelon and interspecific squash.

Color is a parameter used in vegetable production as a quality indicator [29]. A number of differences in color parameters were aroused by colorimetry even though most seedlings had a visibly similar dark green color. A darker color was found under T0 which can be associated with the greater chlorophyll concentration. Generally, increasing B light tends to induce chlorophyll formation and accumulation in watermelon. Fan et al. [30] found greater chlorophyll content for Chinese cabbage seedlings grown under a 6:1 RB, while R developed seedlings with the least chlorophylls. The authors reported that four biosynthetic intermediates (5-aminolevulinic acid, protoporphyrin IX, Mg-protoporphyrin IX, and proto chlorophyll) were decreased under R light, whereas B light seemed to reverse the response. Moreover, Snowden et al. [31] reported significantly increased chlorophyll concentration with increasing B light in seedlings of cucumber, tomato, radish, and pepper grown under 500 μmol m$^{-2}$ s$^{-1}$, and only for tomato seedlings in PPFD of 200 μmol m$^{-2}$ s$^{-1}$.

Both for watermelon and interspecific squash, shoot length was enhanced under the influence of T1 and T3. The two light treatments have the highest FR emission among all lights and the lowest R:FR ratio among the LEDs used in the experiment. The phytochrome photoreceptor senses the R:FR ratio and interconverts between two forms, Pr which is inactive and absorbs R light, and Pfr which is active and absorbs FR light. A higher proportion of FR light leads to a greater conversion of Pfr to Pr, and therefore a decrease of Pfr/Ptotal ratio. This subsequently leads to a number of shade avoidance responses including stem elongation and leaf expansion [19]. T0 and T4 have the greatest B light emission among the tested light sources, which has been found to inhibit internode elongation through the cryptochrome photoreceptors [17] leading to the production of shorter seedlings. Moreover, a high R:FR ratio along with relatively high B light act synergistically and suppress stem elongation to a greater extent than R:FR ratio or B light alone [32]. In cucumber seedlings grown under various RB combinations, greater height and larger leaves were formed with decreasing B light [24]. Moreover, greater height was found in lettuce, spinach, kale, and basil under R light compared to RB [33], whereas a R:FR ratio of 1.5 led to greater plant height in soybean compared to a R:FR ratio of 5 [34].

Leaf area and stem diameter are considered as very efficient indicators of watermelon and interspecific squash seedlings' quality [5]. In our case, T1 and T3 light sources enhanced the leaf expansion and stem thickness in both tested species. Similarly to the shoot length, the two parameters increased with decreasing B light. T3 and T2 have similar B light emission but the high R:FR ratio of the latter led to decreased values in both parameters. Reduction of leaf size under higher B light has previously been reported for cucumber plants grown under FL, HPS or an artificial sun-like spectrum [35]. Moreover, greater leaf area was also found for strawberry plants grown under R light compared to B light [36]. Regarding stem thickness, Li et al. [37] found that a R:B proportion of 3:1 increased stem diameter in tomato compared to monochromatic LEDs.

Similar trends were also observed for the overground dry biomass accumulation where T1 and T3 excelled compared to the rest of the treatments. In general, a light environment rich in R and B wavelengths is beneficial for photosynthesis since these wavelengths are the most photosynthetically efficient according to McCree's [38] relative quantum efficiency curve. Since we only employed light sources emitting both R and B lights, the arisen biomass differences may be highly attributed to the leaf area development which followed a comparable trend. For the relatively low PPFD of

our study, increasing B light acting through the cryptochrome photoreceptors led to smaller leaves. This subsequently reduced the biomass production and accumulation of the seedlings by diminishing the incident light interception. Nanzin et al. [33] reported that lettuce, spinach, and basil had greater dry mass accumulation under a RB with 9% B light, similar to our T1 LED which emits 8% B light. Moreover, additional FR light to RB (R:FR ratio of 3–5 units) led to increased shoot dry weight in two lettuce and one basil cultivar [39].

In both cucurbits tested, the root system development was not significantly affected among the LED light treatments. However, all LEDs enhanced the root dry biomass production compared to T0 which had the lowest R:FR and R:B ratios. The root system must have the potential to absorb sufficient water and nutrients in order to supply the scion during the critical healing period as well as for the rest of its lifetime. Especially for interspecific squash, which is used as rootstock material during grafting, a vigorous, well-developed root system is essential for the production of healthy, high-quality grafted seedlings. FR addition to RB light (R:FR ratio of 3–5 units) has been reported to increase root dry weight of one lettuce and one basil cultivar [39]. In addition, Li et al. [35] found greater tomato root weight under a 3:1 RB combination compared to 1:1 RB or monochromatic LEDs. Moreover, the significantly low dry weight of shoots found in T0 grown watermelon seedlings led to a greater R/S ratio compared to all LED treatments.

Two valuable seedling quality indicators, DW/L and DQI, clearly showed that T0 lamps promoted the production of relatively poor seedlings in both species. The two calculated indices incorporated values of parameters (e.g., shoot dry weight and stem length) that accumulatively excelled under T1 and T3 for watermelon and under T1, T3, and T4 for interspecific squash, subsequently leading to greater values and seedlings of greater quality. DW/L and DQI proved valuable for the evaluation of grafted watermelon seedlings' quality [6].

## 5. Conclusions

In watermelon and interspecific squash seedlings most growth and morphological parameters were enhanced under the influence of T1 and T3, two LEDs emitting relatively high R (53%–64%), and relatively low B (8%–14%), with the least R:FR ratio compared to the rest of the LED treatments. On the contrary, T0 and T4 emitting the highest B percentage led to inferior development and significantly more compact seedlings, while their high G percentage did not contribute to the seedling growth. T2 has a similar spectral emission to T3 but higher R:FR ratio, leading to seedlings of lower quality which were comparable to T0 and T4. The appreciable differences between T3 and T2 prove that even small interplays in the light spectra might trigger a significant response in plant behavior. Generally, a light source with high R emission and additionally rather low B and FR emissions seems ideal for the production of high-quality watermelon (scion) and interspecific squash (rootstock) seedlings.

**Author Contributions:** Conceptualization, methodology, and data analysis: F.B., A.K., A.S.S., and K.R.; experimental measurements: F.B. and C.D.; writing—original draft preparation: F.B. and A.K.; writing—review and editing: F.B., A.K, A.S.S., K.R., and C.D.; supervision and project administration: A.K.

**Funding:** This research was co-financed by the European Union and Greek national funds through the Operational Program Competitiveness, Entrepreneurship and Innovation, under the call RESEARCH– CREATE–INNOVATE (project code: T1EDK-00960, LEDWAR.gr).

**Acknowledgments:** The authors wish to thank Mariangela Fotelli for her help during the experiment.

**Conflicts of Interest:** The authors declare no conflicts of interest. The funding sponsors had no role in the design of the study; in the collection, analyses, or interpretation of data; in the writing of the manuscript; or in the decision to publish the results.

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
