# Peer review of "Optimal LED Wavelength Composition for the Production of High-Quality Watermelon and Interspecific Squash Seedlings Used for Grafting"

_agronomy, doi:10.3390/agronomy9120870_

Round 1

Reviewer 1 Report

I'm very satisfied with their answer. My only suggestion for right now would be that they cite their research to justify using PPFD of 85 μmol m-2s-1 in this research.

Author Response

Thank you very much for your comments and suggestions.

A citation of our research was added in lines 134-136.

Reviewer 2 Report

I have gone through the revised MS. Thank you for agreeing with my comments and acting accordingly!

Author Response

Thank you very much for your comments and suggestions.

Reviewer 3 Report

After the first round of revision the manuscript has been improved. There are still details can be improved. Such as author only added pictures of squash seedlings. If there are pictures of watermelon seedlings, I think it is better to add them.

Author Response

Thank you very much for your comments and suggestions.

A picture of watermelon seedlings has been added in the manuscript (lines 207-208).

This manuscript is a resubmission of an earlier submission. The following is a list of the peer review reports and author responses from that submission.

Round 1

Reviewer 1 Report

Well done! I have enjoyed reading the manuscript and found it very informative. 

Reviewer 3 Report

The objective of the research was to identify the 'optimal'  spectral emission of LEDs with wide wavelength for the production of high quality watermelon scions and interspecific squash rootstocks in a growth chamber. However the PPFD used in the research was only 85±5 μmol m-2s-1. It was so low that the quality of plants in control were all 'lowest not acceptable' or 'lowest acceptable' (Fig1). Although LED treatments had better results than the control, majority of LED treatments for watermelon seedlings was 'lowest not acceptable'. I believe that the low PPFD was the main restricting factor here and results based on such low PPFD made such research NOT usable for field application, and thus the research was set up for failure from the experiment designing stage. 
